# Translation velocity determines the efficacy of engineered suppressor tRNAs on pathogenic nonsense mutations

Nikhil Bharti [1,4], Leonardo Santos[1,4], Marcos Davyt[1], Stine Behrmann[1], Marie Eichholtz[1], Alejandro Jimenez-Sanchez [1], Jeong S. Hong[2,3], Andras Rab [2,3], Eric J. Sorscher [2,3], Suki Albers [1] ✉ & Zoya Ignatova [1] ✉

Nonsense mutations – the underlying cause of approximately 11% of all genetic diseases – prematurely terminate protein synthesis by mutating a sense codon to a premature stop or termination codon (PTC). An emerging therapeutic strategy to suppress nonsense defects is to engineer sense-codon decoding tRNAs to readthrough and restore translation at PTCs. However, the readthrough efficiency of the engineered suppressor tRNAs (sup-tRNAs) largely varies in a tissue- and sequence context-dependent manner and has not yet yielded optimal clinical efficacy for many nonsense mutations. Here, we systematically analyze the suppression efficacy at various pathogenic nonsense mutations. We discover that the translation velocity of the sequence upstream of PTCs modulates the sup-tRNA readthrough efficacy. The PTCs most refractory to suppression are embedded in a sequence context translated with an abrupt reversal of the translation speed leading to ribosomal collisions. Moreover, modeling translation velocity using Ribo-seq data can accurately predict the suppression efficacy at PTCs. These results reveal previously unknown molecular signatures contributing to genotype-phenotype relationships and treatment-response heterogeneity, and provide the framework for the development of personalized tRNA-based gene therapies.

Nonsense mutation-linked genetic disorders are associated with loss-of-function disease phenotypes and severe clinical manifestations. Within the DNA sequence of a protein-coding gene, a nonsense mutation converts a sense triplet encoding an amino acid into one of the three stop codons (UGA, UAA, or UAG), which trigger premature termination of protein synthesis and liberation of the incomplete peptide. In total, 18 out of the 61 sense codons can be converted into premature termination codons (PTCs). Considering the three common PTC identities (UGA, UAA and UAG), nonsense mutations collectively represent the most common type of disease-causing mutations in the human population[1]. Efforts to develop therapies for illnesses

associated with nonsense mutations focus on approaches to miscode the PTC (i.e., to readthrough stop codons) and restore protein synthesis, with either low-molecular-weight pharmacological compounds[2–4] or suppressor tRNAs (sup-tRNAs)[5–9]. However, except for ataluren treatment of nonsense mutations in dystrophin-associated with Duchene muscular dystrophy[2,10] (which has received conditional authorization in Europe), no PTC suppression therapy has yielded an optimal combination of clinical efficacy and safety. A large group of patients with genetic diseases, therefore, remain without treatment.

Low-molecular-weight compounds, such as aminoglycosides, can induce miscoding at PTCs by promoting the binding of a near cognate

[1]Institute of Biochemistry and Molecular Biology, University of Hamburg, 20146 Hamburg, Germany. [2]Department of Pediatrics, School of Medicine, Emory University, Atlanta, GA 30322, USA. [3]Children's Healthcare of Atlanta, Atlanta, GA 30322, USA. [4]These authors contributed equally: Nikhil Bharti, Leonardo Santos. ✉e-mail: suki.albers@uni-hamburg.de; zoya.ignatova@uni-hamburg.de

tRNA, and thus alter the amino acid identity at the affected codon[11]. Accordingly, the PTC identity and nucleotide sequence context surrounding each nonsense mutation influences both readthrough efficacy and identity of the inserted amino acid[4]. Unlike dystrophin, which represents an exception and more readily tolerates amino acid misincorporations[12], many disease-associated proteins are sensitive to miscoding[13–15], thus limiting clinical development and use of small molecule-based therapeutics. By contrast, engineered sup-tRNAs introduce a specific amino acid, which is identical to the affected amino acid[5,16], and exhibit a high molecular safety profile, i.e., with marginal to absent off-target effects at native stop codons (NSC)[5,7,9]. Structure-based mechanistic discoveries of termination at NSCs in eukaryotes[17,18] and of sup-tRNA decoding[16], along with mounting evidence regarding the idiosyncratic structure-function relationship of tRNAs[19–22] have guided sup-tRNA design to achieve high efficacy of targeted PTC suppression. In a previous work, adhering to these design principles, we engineered potent sup-tRNAs suitable for cytosolic delivery encapsulated in lipid nanoparticles. The sup-tRNAs demonstrated remarkable efficacy and molecular safety profiles in vivo, in murine models[5]. Notably, for a specific nonsense mutation (R1162X) associated with cystic fibrosis (CF) pathology, our sup-tRNA achieved efficacy surpassing the projected therapeutic threshold for CF. Sup-tRNA-based therapeutics offer a universal treatment concept: sup-tRNA$^{Arg}_{UCA}$ can suppress UGA PTCs at any affected arginine-encoding (AGA or CGA) codon across multiple disease-associated transcripts. However, we noted that the efficacy of sup-tRNAs is quite variable even at identical PTCs[5]. Despite recent and extensive advances in sup-tRNAs and their importance for clinical applications, a comprehensive understanding of the constraints dictating suppression efficiency at various PTCs has remained elusive.

In this work, we address the differences in the readthrough efficiency of sup-tRNAs by systematically analyzing the mRNA sequence parameters that affect suppression efficacy at various pathogenic PTCs. The short sequence context of PTCs most refractory to suppression did not resemble the sequence context flanking NSCs that establish faithful termination[23–29]. We show that ribosome velocity within a longer sequence stretch upstream of the PTC predominantly determines sup-tRNA efficiency. These findings reveal an unanticipated molecular signature contributing to the PTC rescue efficacy, offering a rationale for treatment strategies customized to the intrinsic susceptibility of specific PTCs to suppression.

## Results

### Short PTC sequence context affects sup-tRNA efficacy only marginally

To address PTC sequence-specific effects on sup-tRNA-mediated readthrough, we designed experiments with only one variable parameter; namely, the identity and sequence context of pathogenic PTCs, while keeping all other experimental variables (e.g., sup-tRNA identity, expression reporters, and cellular system) constant. We considered one very potent sup-tRNA variant, which we engineered previously to incorporate Ser at PTCs (tSA2T5 from ref.[5]). The anticodon sequence of the sup-tRNA$^{Ser}$ was adjusted to decode each of the three PTC identities—for UGA we used sup-tRNA$^{Ser}_{UCA}$, for UAG—sup-tRNA$^{Ser}_{CUA}$, whereas for UAA—sup-tRNA$^{Ser}_{UUA}$. In vitro transcribed (IVT) sup-tRNAs were co-transfected in human bronchial epithelial cells, CFBE41o⁻, with a plasmid-encoded PTC reporter of firefly luciferase (FLuc, Fig. 1a); the coding sequence of the FLuc reporter was extended at its 5′ end by 15 codons representing the short sequence context of various pathogenic PTCs in three disease-associated genes (i.e. the cystic fibrosis transmembrane conductance regulator (CFTR) linked to CF or in dynein axonemal heavy chain 5 (DNAH5) or radial spoke head component 4 (RSPH4A) implicated in primary ciliary dyskinesia (PCD) (Supplementary Tables 1, 2)).

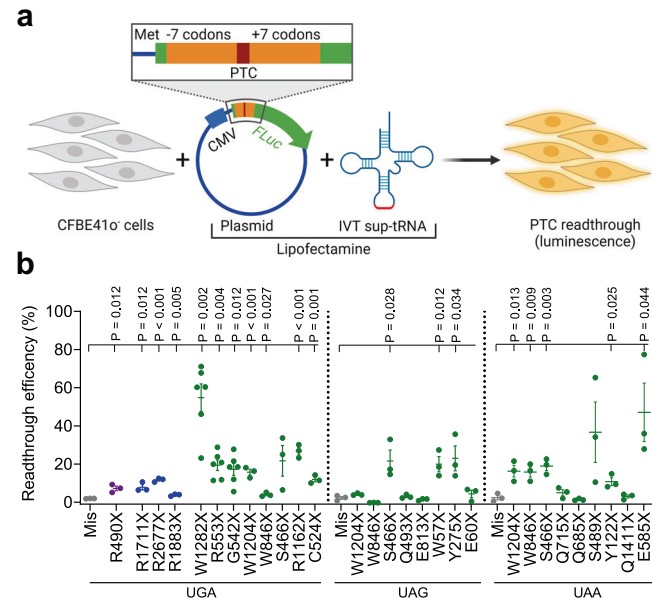

**Fig. 1 | Efficiency of sup-tRNA on PTCs flanked by their natural short-sequence context. a** Schematic of the readthrough experiment with co-transfected IVT sup-tRNA and plasmid-encoded PTC-FLuc variants. IVT in vitro transcribed sup-tRNA, FLuc (green) firefly luciferase, CMV cytomegalovirus promoter, Met methionine start codon of FLuc, PTC (red) premature termination codon. Each pathogenic PTC is flanked by the natural sequence context (seven codons upstream and seven codons downstream, orange) of the respective PTC. Figure was generated using BioRender. **b** Readthrough efficiency of sup-tRNA$^{Ser}_{UCA}$ decoding UGA, sup-tRNA$^{Ser}_{CUA}$ pairing to UAG, and sup-tRNA$^{Ser}_{UUA}$ decoding UAA at PTC mutations in RSPH4A (purple) and DNAH5 (blue) and in CFTR (green) tested in CFBE41o⁻ cells and normalized to the expression of wildtype FLuc. Mis, mismatched tRNA (gray) that does not pair to any of the PTCs. Data are means ± s.e.m. (n = 3 independent replicates for all mutations except n = 6 for G542X, R553X, and W1282X). Statistics, two-sided t test Source data are provided as a Source Data file.

This cognate sup-tRNA$^{Ser}$ suppressed all three PTCs (Fig. 1b), including the UAA PTC which is the highest fidelity termination codon[30] and most difficult to correct by aminoglycosides[4] or sup-tRNAs[5]. For both UGA and UAA PTCs, the readthrough varied considerably, i.e., from 4% for R1883X in DNAH5 and W846X in CFTR to 55% for W1282X in CFTR (Fig. 1b). The variations in the readthrough efficiency were smaller for UAG PTCs (Fig. 1b). Notably, for some mutations (e.g., W1204X, W846X) despite the same sequence context, the PTC identity substantially affected the readthrough efficacy, whereas, for the S466X PTC, we detected relatively similar readthrough at all three (UAA, UGA, and UAG) PTCs. Overall, these data imply a strong sequence context-dependent suppression efficacy—an interplay between the PTC identity and the sequences flanking each PTC.

The identity of NSCs and surrounding sequence contexts influence the probability and fidelity of termination[4,23–31]. By analyzing the available literature, we extracted a conserved signature of sequences upstream and downstream of NSCs that greatly facilitate readthrough, and assembled an artificial sequence with potentially high readthrough efficacy, named underlined efficient readthrough context (ERC, Fig. 2a). The sequences upstream or downstream of the PTCs in CFTR, DNAH5, and RSPH4A exhibited different degrees of homology to the ERC (Fig. 2b and Supplementary Fig. 1a). We observed almost no correlation between the degree of sequence homology to ERC and the readthrough efficiency: PTCs with nearly no similarity to the upstream ERC sequence (e.g., S489X, E585X, R1162X, Y122X, and Q715X) showed comparatively high readthrough efficiency, while for some PTCs (e.g., W57X, G542X, R553X, W846X, W1282X, Q1411X) with high context

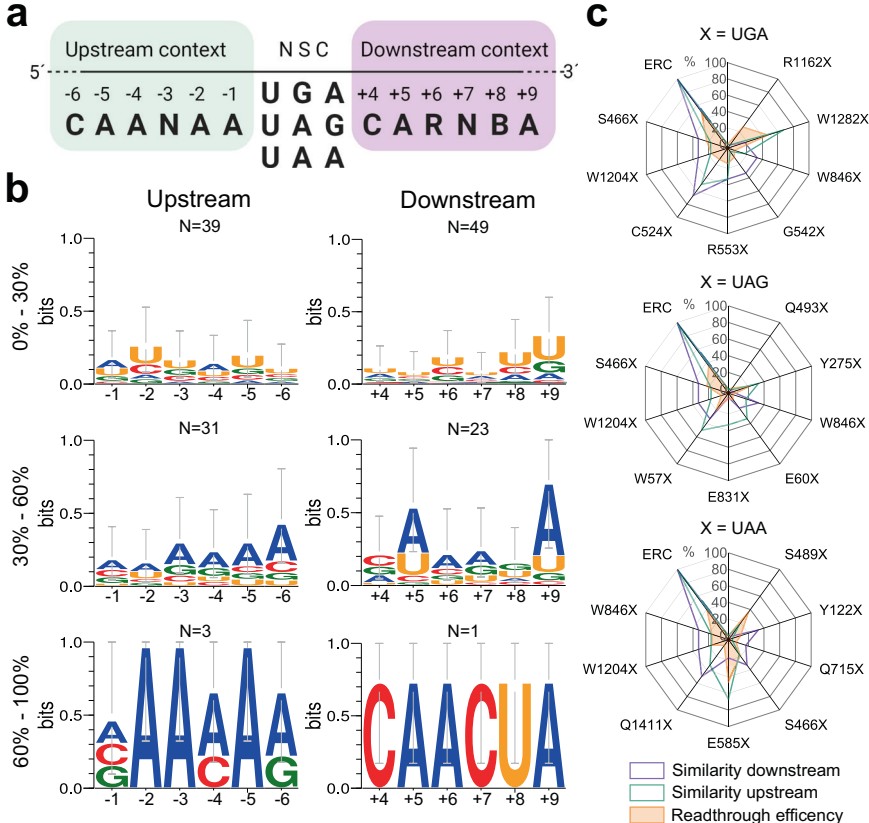

**Fig. 2 | Homology to the ERC does not correlate with PTC suppression efficiency. a** Sequence identity upstream (green) and downstream (purple) of the efficient readthrough context (ERC) maintaining the highest readthrough at native stop codons (NSCs). $R = A/G$; $B = U/C/G$; $N = A/U/G/C$. The first nucleotide of NSC is numbered +1. Figure was generated using BioRender. **b** Sequence logo of the 6 nt-sequence upstream (left plots) or downstream (right plots) of all PTCs in *CFTR*. Sequences were assigned in three groups based on their resemblance to the ERC (percentage on the left). *N*, number of PTC sequences in each group. **c** Spider diagram for various PTCs in *CFTR* displaying the experimentally measured suptRNA readthrough efficiency (orange filled, Fig. 1b) and the resemblance to the upstream (green line) and downstream (purple line) sequence of the ERC. The readthrough efficiency at the ERC was 50% (at ERC with UGA), 39% (at ERC with UAG), and 35% (at ERC with UAA). Source data are provided as a Source Data file.

similarity to the downstream ERC sequence, the readthrough efficiency largely varied (Fig. 2c and Supplementary Fig. 1b).

### Translation velocity upstream of the PTC modulates sup-tRNA efficacy

In a previous study, we noted that the sup-tRNA efficacy differs within the full-length transcript, in comparison to decoding a PTC embedded in a short sequence context[5]. Thus, we next addressed the ability of sup-tRNA[Ser] to restore the expression of a full-length protein. In a pilot experiment, we co-transfected this sup-tRNA with a plasmid-encoded *PTC-CFTR* variant (*R1162X-CFTR*) in two different laboratory cell lines, HEK293 and Hep3B. In the context of the full-length *CFTR* transcript, the sup-tRNA[Ser] yielded 2- and 6-fold higher protein in HEK293 and Hep3B cells, respectively, compared to the background readthrough with a mismatched tRNA (Supplementary Fig. 2a). This is in contrast to the noticeably lower readthrough activity at R1162X PTC when shorter PTC context was tested (Fig. 1b), implying that a larger PTC sequence context likely affects sup-tRNA efficacy. Since CFTR is sensitive to amino acid misincorporations[15] and the sup-tRNA[Ser]-mediated Ser misincorporation may alter protein stability, we also compared the efficacy of sup-tRNA[Arg] (tRT5[5]) at R1162X PTC in different cells that express *R1162X-CFTR* exogenously from a plasmid cDNA (CFBE41o- cells) or endogenously with all introns and exons (16HBEge R1162X/- and hNE R1162X/R1162X). Among different cell types, CFTR expression levels differed by 6–10 fold (Supplementary Fig. 2b). One explanation for distinctive readthrough efficiencies in various cells, independent of the transcript architecture, is a cell-specific compositional variation of

the translation apparatus. The most striking difference in this respect is the tRNA abundance[32–36]—a key determinant of the codon translation velocity (reviewed also in ref. 37). Thus, we reasoned that the mRNA translation velocity, particularly, in the upstream proximity of PTCs, may influence sup-tRNA-mediated readthrough. Using Ribo-seq (or ribosome profiling) from CFBE41o- cells, we determined the ribosomal dwelling occupancy at A-site codons following calibration of the ribosome-protected fragments at the ribosomal A site (i.e., the site accepting the aminoacyl-tRNA). The reciprocal value of the ribosomal occupancy aggregated across all translated transcripts is an estimate of the codon speed and inversely correlates with the concentration of its cognate tRNA[37–40]. We next used these values to compute the average translation speed of all possible consecutive 5-codon sequences across the entire transcriptome. This metric follows largely a Gaussian distribution, that is, many of the 5-codon stretches were translated at the same average speed (Supplementary Fig. 3a). Computing the mRNA translation velocity within 70 codons upstream of PTCs (Fig. 3a) revealed that for many PTCs, the average translation velocity was within a $1.5\sigma$ interval (Supplementary Fig. 3b, c). Under normal growth conditions, translating ribosomes are spatially separated, with an average inter-ribosomal distance (IRD) between the A sites of two consecutive ribosomes of ~66 codons[38]. We observed that some regions were translated more rapidly or slower than the velocity within the $1.5\sigma$ interval (Fig. 3a). Strikingly, for many PTCs, we detected that fast-translating regions were followed by slow-translating regions (or vice versa), thus causing an abrupt change of the translation velocity within consecutive segments (Fig. 3b and Supplementary

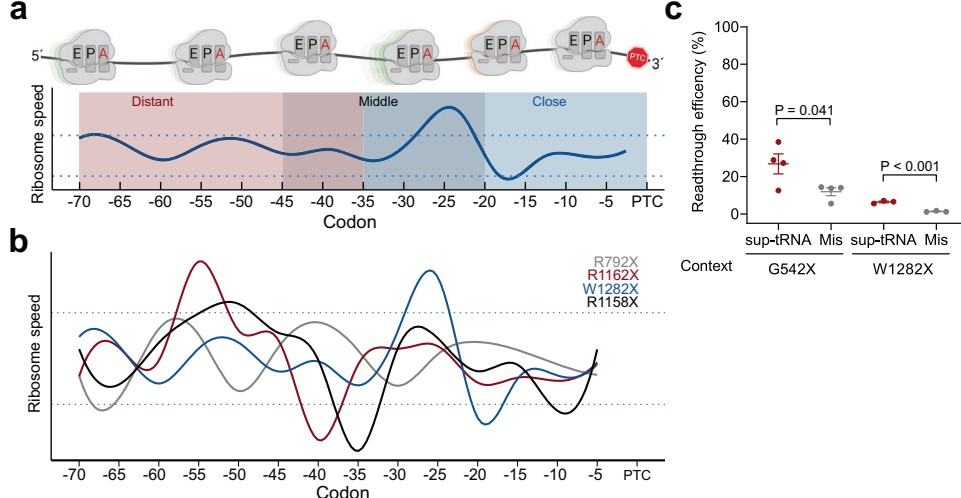

**Fig. 3 | mRNA translation velocity modulates PTC suppression susceptibility.**
**a** Computed translation velocity of the region upstream of the PTCs. Codon speed was estimated by Ribo-seq from CFBE41o⁻ cells. The curves are smoothed with a window of five codons. Shades represent the three groups with speed inversions categorized dependent on the distance to the PTCs: close (5–35 codons, blue), middle (20–45 codons, shaded), and distant (35–70 codons, red). The cartoon above the plot was generated using BioRender. **b** Examples of translation profiles of PTCs in *CFTR* (for all PTCs—see Supplementary Fig. 3b). Horizontal dotted lines designate ±1.5σ (Supplementary Fig. 3a). PTC with a smooth translation profile (i.e., within 1.5σ) (gray); PTCs with inversion of the translation speed (i.e., >1.5σ) and related to the distance to PTC as close (blue, 5–35 codons), middle (black, 20–45 codons) and distant (red, 35–70 codons) are shown. **c** Readthrough efficiency of sup-tRNA (red) at G542X and W1282X PTC in full-length CFTR PTC tested in CFBE41o⁻ cells. Mis, mismatched tRNA (gray) that does not pair to PTC. Data are means ± s.e.m. (*n* = 3 independent replicates for W1282X and *n* = 4 for G542X). Statistics, two-sided *t* test. Source data are provided as a Source Data file.

Fig. 3b, c). Dependent on the distance between these speed inversions and individual PTCs, we categorized them into three groups: within (i) close (1–35 codons), (ii) middle (20–45 codons), or (iii) distant (35–70 codons) regions upstream of the PTC (Fig. 3a, b and Supplementary Fig. 3b, c).

In group (i) with translation velocity changes proximal to the PTC are two mutations, G542X and W1282X (Fig. 3b), which are among the most refractory to suppression and for which aminoglycosides are ineffective in rescuing full-length CFTR protein[41]. By contrast, in our luciferase reporter assays with short sequence contexts, the sup-tRNA readthrough efficiency at both PTCs was robust, e.g., 20% for G542X and 55% for W1282X, with W1282X exhibiting the highest readthrough among all tested PTCs (Fig. 1b). We reasoned that readthrough at a PTC within the full-length transcript may differ (compare Fig.1b and Supplementary Fig. 2a). In our previous work[5], we also observed striking distinction between suppression efficacy within the full-length transcript in comparison to decoding a PTC embedded in a short sequence context (e.g., S466X suppression in a reporter construct is ~20%, while embedded in the full-length S466X-CFTR increases to ~70%[5]). Thus, we next measured suppression at G542X and W1282X PTCs within the full-length *CFTR* transcript and monitored the readthrough by assessing the full-length CFTR protein with western blotting. In contrast to the efficacy within the short sequence contexts (Fig. 1b), the suppression efficacy at the same PTC within the full-length transcript was reduced down to only a few percent above the mismatched tRNA (Fig. 3c), suggesting that a larger sequence context determines suppression efficacy at PTCs.

To address the causality between the velocity change and the sup-tRNA efficacy, we extended the *FLuc* sequence with 35 codons preceeding W1282X PTC (eW1282X PTC; Fig. 4a, b and Supplementary Table 3). Through synonymous substitutions (i.e., changes in the nucleotide sequence without changing the encoded amino acids), we next produced a variant (smW1282X PTC) with smoothed translation profile, so that the velocity fluctuations fell within the 1.5σ interval (Fig. 4b). Notably, the readthrough efficiency at the smW1282X PTC increased by 9% (Fig. 4c), implying that reversal of the translation speed negatively affected sup-tRNA-mediated PTC suppression.

In the absence of sup-tRNA, decoding of PTCs is mediated—similarly to NSCs—by the eukaryotic release factor (eRF1), a tRNA-shaped entity stimulated by the GTPase eukaryotic release factor 3 (eRF3)[18]. To eliminate competition between eRF1 and sup-tRNA at the PTC, we added the inhibitor SRI-41315 (5 μM) of eRF1[3], and reasoned that lack of competition would potentiate sup-tRNA-mediated readthrough. Notably, the inhibition of the eRF1 resulted in higher sup-tRNA readthrough only at smW1282X PTC but not at eW1282X PTC (Fig. 4c), suggesting that competition with eRF1 is not the sole factor that antagonized sup-tRNA efficacy at eW1282X PTC.

## Ribosomal collisions at PTC negatively affect sup-tRNA efficiency

The abrupt translation velocity change within the W1282X upstream sequence, i.e., from fast (codons 29–24) to slow (codons 20–17), would be expected to shorten the IRD between the trailing and leading ribosomes at the W1282X PTC so that ribosomal collisions might occur. To probe for collisions, we utilized a Calu3 cell line, which was manipulated to endogenously express equal amounts of *WT-CFTR* or *W1282X-CFTR* transcripts. We isolated poly-ribosomes (polysomes) and treated them with RNaseI to select for genuinely colliding ribosomes[42] which are largely nuclease-resistant and migrate as stable di-ribosomes (disomes) (Supplementary Fig. 4).

We probed the disomes for two collision markers, GCN1 and ZNF598 (Fig. 5a). GCN1 interacts with both adjoining ribosomes and spans from the P-stalk region of the trailing collided ribosome to the P-stalk and the A-site region of the lead ribosome[43]. The ubiquitin ligase ZNF598 is a direct sensor of ribosome collisions and site-specifically ubiquitinates ribosomal proteins at the 40S-40S interface, which initiates the collision-resolving cascade[44]. Nuclease-resistant disomes from the W1282X-CFTR expressing cells were positive for both GCN1 and ZNF598 (Fig. 5a). The observed reactivity against CFTR antibodies (Fig. 5b) suggests that the majority of these disomes represent *CFTR*-translating collided ribosomes. By contrast, WT-CFTR expressing cells showed a marginal signal for GCN1 and ZNF598 in the RNase-treated disomes (Fig. 5a), that was negative for CFTR (Fig. 5b), and thus, likely represents basal levels of colliding ribosomes observed in human cells[45].

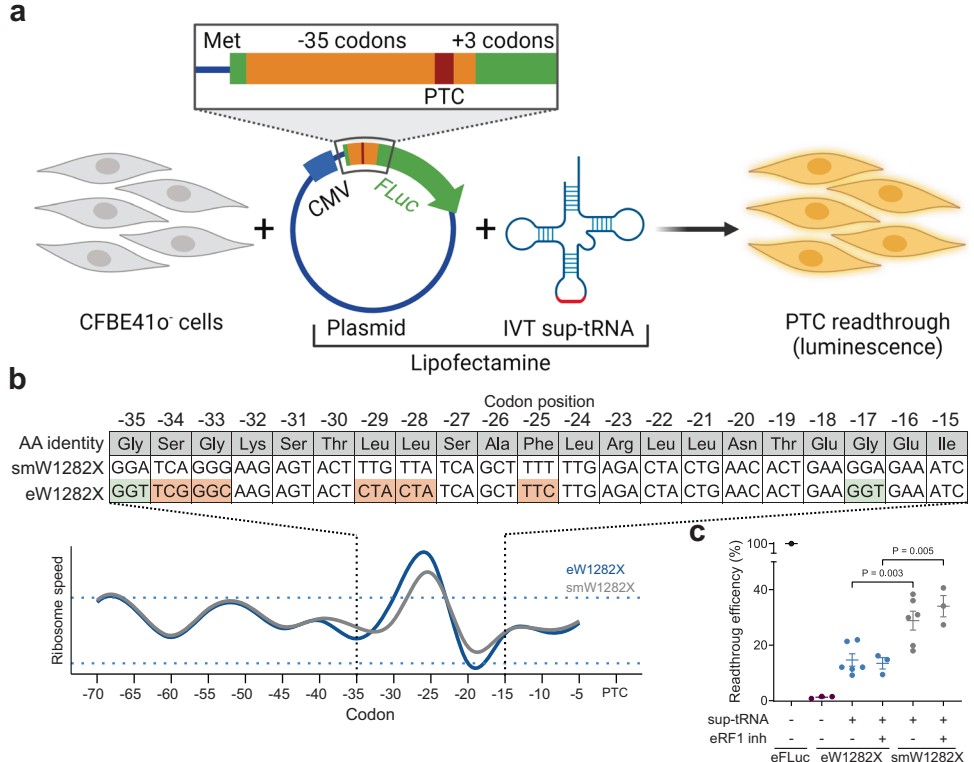

**Fig. 4 | Alteration of the mRNA translation velocity improves PTC suppression.** **a** Schematic of the readthrough experiments with plasmid-encoded *PTC-FLuc* constructs with a longer sequence upstream of the PTC (orange). IVT in vitro transcribed sup-tRNA, *FLuc* (green) firefly luciferase, CMV cytomegalovirus promoter, Met methionine start codon of *FLuc*, PTC (red) premature termination codon. Figure was generated using BioRender. **b** Computed translation velocity profiles of eW1282X PTC with extended sequence upstream of the PTC (blue) and codon-substituted variant smW1282X PTC (gray) of *CFTR* whose translation profile was smoothed to be within 1.5σ. Inset, codon, and amino acid sequence of the segment with speed inversion (light green, substitution for a fast-translated codon; orange, substitution for a slow-translated codon). Horizontal dotted lines designate the ±1.5σ interval. **c** Sup-tRNA mediated readthrough efficacy in CFBE41o⁻ cells, with and without eRF1 inhibitor (SRI-41315, 5 μM) and normalized to wildtype FLuc extended with the same sequence without the PTC (eFLuc). Mis, mismatched tRNA that does not pair to the PTC (purple). Data are means ± s.e.m. (*n*, independent replicates, e.g., *n* = 3 for eFLuc, mismatched tRNA, eW1282X and smW1282X with sup-tRNA and eRF1 inhibitor; *n* = 6 for eW1282X and smW1282X with sup-tRNA). Statistics, two-sided *t* test. Source data are provided as a Source Data file.

Together, these results imply that ribosomal collisions at native W1282X PTCs substantially diminished sup-tRNA-mediated suppression.

## Predictive modeling of collision probability

Termination is ~4 times slower than elongation. In the first step of termination, eRF3 in complex with GTP rapidly delivers eRF1 to the stop codon in the A site with kinetics similar to the rapid association of the elongating complex (eEF1A/tRNA/GTP) with an A-site sense codon[18]. In the decoding center, the sup-tRNA establishes a perfect Watson–Crick geometry[16]. We assume that sup-tRNA decoding velocity at the PTC ($v_{sup}$) will be similar to that of the sense codon decoding, namely ~5.6 codons/s[38]. Consequently, for sequences translated with a uniform speed (i.e., within the 1.5σ interval), two consecutive ribosomes will elongate with an average IRD of ~66 codons (Fig. 6a), reflecting the average distance of elongating ribosomes across the human transcriptome[38]. Sequence-context dependent change of the speed of the trailing ribosome ($v_t$) will alter the average IRD to the leading PTC-decoding ribosome (Fig. 6a), reaching in some cases a critical distance of ten codons, which is the IRD between A sites of two colliding ribosomes[42,44]. We modelled how an abrupt speed change upstream of a PTC will affect the ribosome traffic and the IRD between two consecutive ribosomes (Fig. 6b). Within an interval of 10–100 codons upstream of each PTC, we computed every possible IRD between two consecutive ribosomes, together with translation velocity changes within the interval 0–0.5. The model predicted a higher probability of ribosomal collisions at PTCs with abrupt

translation velocity changes in close proximity to the PTC (x-axis, Fig. 6b) leading to shortening of the IRD between two consecutive ribosomes (y-axis, Fig. 6b). Thus, the greater the velocity change, the more pronounced the impact on the IRD between two consecutive ribosomes and the higher the probability for collisions (red shaded area, Fig. 6b). The model also predicted a weak effect of translation velocity changes when ribosomes are within large IRDs, i.e., irrespective of the magnitude of the velocity change, ribosomes with IRD above an average transcriptomic IRD of 66 codons, are less prone to collide (dashed line Fig. 6b).

To select PTCs whose upstream sequences are prone to trigger collisions, we next used the computed translation profiles of sequences proximal to the PTCs (70 nt upstream; Fig. 3a and Supplementary Fig. 3) and calculated translation velocity changes in genes associated with CF and PCD. With this information, we next modeled the IRDs for each mutation (Fig. 6b and Supplementary Fig. 5). Notably, for PTCs with speed alterations in distant regions (35–70 codons), the model predicts no probability for collisions (triangles; Fig. 6b and Supplementary Fig. 5). In turn, changes of translation velocity in regions close to the PTC (5–35 codons) are predicted to shorten the IRD between the leading ribosome at the corresponding PTC and the trailing ribosome, hence increasing the likelihood of collisions (squares; Fig. 6b and Supplementary Fig. 5). For both G542X and W1282X PTCs, the model predicted high likelihood of collisions (Fig. 6), implying strong predictive power in assessing the susceptibility of non-sense correction with sup-tRNAs.

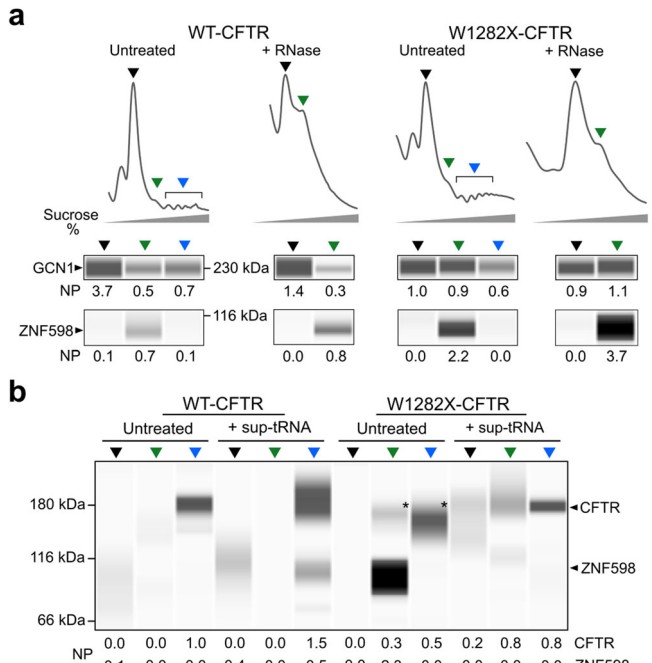

**Fig. 5 | Ribosomal collisions at PTCs decrease sup-tRNA efficacy. a** Nuclease-resistant disomes from Calu3 expressing W1282X-CFTR or WT-CFTR. Different ribosomal fractions (monosomes−black, disomes− green, and polysomes−blue) were separated through sucrose gradients (horizontal gray triangle designates the sucrose gradient) and probed with GCN1 and ZNF598 antibodies. The numbers under the blots denote normalized protein band (NP) normalized to the loaded protein measured as total protein amount. **b** Simultaneous probing with CFTR and ZNF598 antibodies of Calu3 cells expressing WT-CFTR or W1282X-CFTR. Cells expressing W1282X-CFTR and not treated with sup-tRNA (untreated) produced a shorter CFTR variant (marked with *). Treatment with sup-tRNA completely abolished the strong ZNF598 signal. Numbers under the blots denote the normalized protein band (NP) ratioed to the loaded protein measured as total protein amount. These experiments serve as a quality check for disomes (**a**) or collision markers (**b**) and hence, were performed as single experiments. Source data are provided as a Source Data file.

## Discussion

In conclusion, our results reveal that translation velocity within a longer sequence context upstream of a PTC determines the sup-tRNA-mediated suppression efficacy. Speed alterations in close proximity to the PTCs or within a distance with a length 3–4 times of the minimal distance for collisions (<35 codons) are detrimental (Fig. 6a). By contrast, speed alterations in regions far from the leading ribosome (>35 codons) can still be buffered by the remaining distance to traverse, so that overall two consecutive elongating ribosomes will remain spatially separated (Fig. 6a). Abrupt changes in the ribosomal speed proximal to PTCs (<35 codons) diminish the sup-tRNA-mediated suppression efficacy due to a high likelihood to incur ribosomal collision. In such cases, addition of low-dose eRF1/3 inhibitors[3,46–48] as an adjuvant may reduce collisions and augment suppression efficacy. Translation initiation is widely recognized as the most potent factor in determining gene expression[49]. Elevated initiation rates have been associated with diminished protein expression of transcripts containing sites that induce ribosome stalling and collisions[50]. Conversely, inefficient translation initiation may promote the evasion of ribosome collisions[50]. Consequently, an alternative approach to reduce collisions associated with PTCs and enhance sup-tRNA effects could involve introducing a low dose of initiation inhibitors. Translation inhibitors, primarily designed to disrupt the function of eIF2α as a potent regulator of cap-dependent translation, are actively being explored as potential anticancer agents[51,52]. Regardless of the chosen approach,

calibration of adjuvant dosage and treatment must be carefully tuned to avoid widespread misregulation of translation initiation or termination.

For PTCs with a similar sequence-dependent translation signature but from different mRNAs, the degree of collisions may differ. For example, a PTC within a highly translated transcript at high initiation frequency and consequently shorter average IRD would be more likely to produce collisions, in contrast to a similar PTC embedded within a less robustly translated mRNA. From that perspective, several factors (e.g. 5′UTRs that modulate initiation, ribosome loading decay for each transcript) emerge as potent determinants of collision likelihood[53–55]. Similarly, because of compositional variations of the translational households, including cell type- and tissue-specific differences in the initiation rates and the mRNA utilization, will consequently reflect the degree of the ribosome collisions at otherwise identical PTCs. Incorporating these additional parameters into further development of the model presented here is expected to augment its predictive accuracy. The likelihood of strong, cell type-specific PTC suppression implies that Ribo-seq data or tRNA levels−if easily accessible−from disease tissues would be the precise predictor of translation signature and suppression efficacy, thus providing a framework for clinical trial design and development of personalized sup-tRNA-based therapies of nonsense mutations.

## Methods

### Cell lines, reporter constructs, and tRNA transcription

To test the readthrough efficiency of sup-tRNA, we used firefly luciferase (FLuc) as a reporter. A stretch of 15 codons (45 nts) from distinct disease-related genes centered at the respective PTC mutations (Supplementary Table 2) was inserted into pGL4.51 (Promega) harboring the *luc2* gene, with the 5′ *luc2* coding sequence (CDS), directly downstream of the ATG start codon, yielding PTC-FLuc variants. In addition, the CDS of *luc* was extended 5′ upstream, in-frame (after the Luc ATG start codon) by longer context for W1282X (i.e., 35 codons upstream and 3 codons downstream of the W1282X PTCs) (Supplementary Table 3).

HEK293 (CRL-1573) and Hep3B (HB-8064) cell lines were obtained from the ATCC. The immortalized CF bronchial epithelial cell line (CFBE41o⁻; generated by D. Gruenert, UCSF) with no detected allelic CFTR expression was used for ectopic expression of CFTR variants. Calu3 epithelial cells isolated from an individual with lung adenocarcinoma were purchased from ATCC (HTB-55). The Calu3 cells endogenously express *CFTR* (three *CFTR* loci per cell), and cells were clonally expanded and karyotyped to obtain a homogeneous parental cell population. Cells were transfected iteratively with zinc finger nuclease and donor DNA reagents carrying the W1282X PTC. Single-cell clones were established through a process of serial pooling. Successful gene editing was validated with next-generation sequencing, mutation-specific quantitative PCR, and W1282X-directed ddPCR. We identified a clonal Calu3 line in which all three wild-type *CFTR* copies were successfully modified to obtain the W1282X variant. A ubiquitous chromatin opening element was further integrated into one *CFTR* copy resulting in an increase of *W1282X-CFTR* mRNA to levels comparable to wildtype *CFTR* mRNA in the parental Calu3 cells.

16HBE14o- cells were gene edited at the endogenous CFTR locus using CRISPR/Cas9 to establish isogenic 16HBEge CFTR R1162X/- cell lines[56]. The 16HBEge model was obtained from the Cystic Fibrosis Foundation Therapeutics Lab (Lexington MA, USA). Primary human nasal epithelial (hNE) cells collected by nasal brush from a CF patient homozygous for R1162X/R1162X were obtained at passage 2 from the Cystic Fibrosis Foundation Therapeutics Lab (Lexington MA, USA).

Calu3, HEK293 CFBE41o⁻ cells or 16HBEge CFTR R1162X/- cells were cultured in Minimum Essential Medium (MEM, Pan Biotech) supplemented with 10% fetal bovine serum (FBS, Pan Biotech) and 2 mM ʟ-glutamine (Thermo Fisher Scientific), for Hep3B Dulbecco's

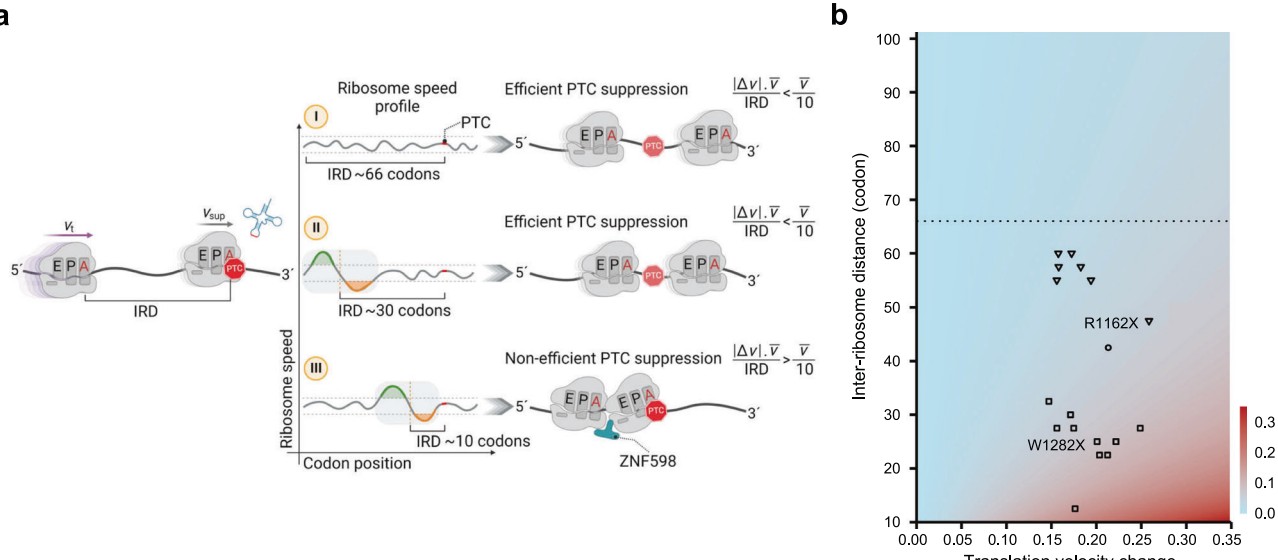

**Fig. 6 | Translation velocity profile determines the PTC suppression efficacy and collision probability. a** Schematic of the different models of PTC suppression. The speed of the trailing ribosome ($v_t$) and leading ribosome with the sup-tRNA decoding the PTC ($v_{sup}$) determine the inter-ribosomal distance (IRD). (I) By sequence context with a smooth translation profile (i.e., within $1.5\sigma$), the speed of both ribosomes are nearly equal ($v_t$–$v_{sup}$) and also equal to the average speed of translation ($\bar{v}$), with an IRD of ~66 codons, i.e., comparable to that observed transcriptome-wide[38]. An inversion of the translation speed (i.e., $>1.5\sigma$) that is distant (35–70 codons) (II) or close (5–35 codons) (III) to the PTC allows the trailing ribosome to gain speed and shorten the distance to the leading ribosome. At the critical distance of ten codons (III) ribosomes collide and collisions are sensed by ZNF598. $\Delta v$ represents changes in translation velocity (green and orange marked areas). Figure was generated using BioRender. **b** Modelling collision probability at PTCs depends on the degree of translation velocity change in sequences upstream of the PTC (x-axis) and the IRD between the leading and trailing ribosomes (y-axes). The horizontal dashed line denotes the IRD of 66 codons. PTCs in CFTR are categorized by velocity changes related to the distance to PTCs (profiles shown in Fig. 3b and Supplementary Fig. 3b): close (5–35 codons, squares), middle (20–45 codons, circle) and distant (35–70 codons, triangles). The identity of the mutations is provided in the Source Data file.

Modified Eagle's Medium (DMEM, Pan Biotech) supplemented with 10% FBS (Pan Biotech) and 2 mM L-glutamine (Thermo Fisher Scientific). CF patient-derived hNE cells homozygous for R1162X mutation, grown at the air-liquid interface and differentiated into ciliated pseudostratified epithelial monolayer, were transfected with tRT5 or mismatched tRNA[5]. Cells were expanded for 3–5 days until they reached 70–80% confluency in PneumaCult ALI Ex⁺ media (StemCell kit). To 500 mL medium 0.5 mL hydrocortisone (StemCell), 10 mL 50X Ex⁺ supplement (StemCell kit), a 2 mL of amphotericin B (12.5 μg/mL; Sigma), 500 μL ceftazidime (100 mg/mL; Sigma), 500 μL vancomycin (100 mg/mL; Sigma), and 500 μL tobramycin (100 mg/mL; Sigma) were added. Thereafter, cells were detached by 0.05% trypsin-EDTA (Pan-Biotech) and seeded onto 12- or 24-ALI transwells (0.4 μm pore polyethylene terephthalate membrane inserts, Corning) coated with collagen IV at a confluency of $1.5 \times 10^5$–$2 \times 10^5$/well. Complete Ex⁺ media (without antibiotics) was added basolaterally (0.5–0.6 mL) and apically (~0.5 mL). Media was changed every day on each side over the growth of 3–4 days. On day 4, the apical medium was removed, and basolateral bathing solution was exchanged for ALI complete medium (StemCell), followed by additional exchanges three times/week for at least 21 days until reaching a fully differentiated state[5].

Sup-tRNAs or mismatched tRNA were IVT using the T7 transcription system as described[5,16]. Namely, for in vitro tRNA synthesis, two partially overlapping DNA oligonucleotides encoding the corresponding full-length tRNA sequence with an upstream T7 promoter (5′-TAATACGACTCACTATA-3′) were used. 24 μM of both oligonucleotides were denatured (2 min at 95 °C), aligned for 3 min at room temperature in 20 mM Tris-HCl (pH 7.5), thereafter 0.4 mM dNTPs were added and incubated with 4 U/μL RevertAid Reverse Transcriptase (Thermo Fisher Scientific) for 40 min at 37 °C. For in vitro T7-guided transcription, 2 mM NTPs, 5 mM GMP, 1x transcription buffer, 0.6 U/μL T7 RNA polymerase (Thermo Fisher Scientific) were added and incubated overnight at 37 °C. The sup-tRNAs were purified on preparative denaturing polyacrylamide gel electrophoresis (PAGE) by eluting with 50 mM KOAc, 200 mM KCl pH 7.0 overnight at 4 °C, followed by ethanol precipitation and re-suspension in DEPC-$H_2O$.

### tRNA transfection and in vitro luciferase readthrough assay

HEK293, Hep3B, or CFBE41o⁻ cells were seeded in 96-well cell culture plates at $1 \times 10^4$ cells/well and grown in Dulbecco's Modified Essential Medium (DMEM, Pan Biotech or Gibco) for Hep3B or Minimum Essential Medium (MEM, Pan Biotech) for CFBE41o⁻ cells supplemented with 10% fetal bovine serum (FBS, Pan Biotech) and 2 mM L-glutamine (Thermo Fisher Scientific). 24 h later, cells were co-transfected in triplicate with 25 ng PTC-FLuc or WT-FLuc plasmids and 100 ng of IVT sup-tRNA or mismatched tRNA using lipofectamine 3000 (Thermo Fisher Scientific). After 4–6 h, the medium was replaced and 24 h post-transfection cells were lysed with 1x passive lysis buffer (Promega). Luciferase activity was measured with luciferase assay system (Promega) on Spark microplate reader (Tecan).

In some experiments, cells were treated with 5 μM eRF1 inhibitor (SRI-41315, MedChemExpress) and after 18 h transfected with sup-tRNA as described above for 6 h (total duration of eRF1 treatment was 24 h). Thereafter, cells were lysed with 1x passive lysis buffer (Promega). Luciferase activity was measured with luciferase assay system (Promega) on Spark microplate reader (Tecan).

### Polysome profiling

Cells (~2–3 million for each experiment) were harvested in polysome lysis buffer (10 mM Tris-HCl (pH 7.4), 5 mM $MgCl_2$, 100 mM KCl, 1% Triton X-100) supplemented with 2 mM DTT and 100 μg/ml cycloheximide to stabilize ribosomes on mRNA during postprocessing steps. Samples were layered onto a 5 ml sucrose gradient (15–60%) and centrifuged for 1.5 h at $148,900 \times g$ (Beckman Coulter, SW 55Ti rotor) at 4 °C. Gradients were fractionated on a piston gradient fractionator (Biocomp) and absorbance at 254 nm was recorded.

## CFTR immunoblot expression analysis

Fractions from the sucrose gradients, or cells (CFBE41o⁻, HEK293, Hep3B, 16HBEge (R1162X/-), primary human nasal epithelia (hNE; R1162X/R1162X) Calu-3 WT-CFTR, or Calu-3 W1282X-CFTR), transfected with sup-tRNA or mismatched tRNA, were lysed with 80 μL of MNT buffer (10x; 300 mM Tris-HCl pH 7.5, 200 mM 2-(*N*-morpholino) ethanesulfonic acid and 1 M NaCl) and lysates were subjected to immuno-blotting. CFTR was detected with monoclonal anti-CFTR-NBD2 antibody (1:100 dilution, #596, John R. Riordan and Tim Jensen, University of North Carolina, Chapel Hill, USA) available through the Cystic Fibrosis Foundation Therapeutics Antibody Distribution Program (Bethesda, USA Company). Analysis was conducted with an automatized capillary electrophoresis system (JESS, ProteinSimple). The peak area corresponding to fully glycosylated CFTR (band C) was normalized to total protein using the JESS quantification module using the JESS instrument software (Compass for SW Version 6.0.0). ZNF598 was detected with rabbit polyclonal anti-ZNF598 antibody (1:50 dilution, Abcam ab80456. Lot: GR32971 30-2), and GCN1 probed using rabbit anti-GCN1L1 antibody (1:50 dilution, Bethyl Laboratories, cat# A301-843A).

For mutations in full-length CFTR, CFBE41o⁻ cells were seeded in 6-well cell culture plates at $1 \times 10^5$ cells/well and cultured in MEM medium supplemented with 10% FBS and 2 mM L-glutamine (Thermo Fisher Scientific). 16–24 h after seeding, cells were co-transfected with 400 ng plasmid bearing CFTR-PTC (W1282X) and 200 ng of each tRNA using lipofectamine 3000 (Thermo Fisher Scientific). After 4–6 h, the medium was replaced and 24 h post-transfection cells were lysed with 80 μL of MNT buffer (10x; 300 mM Tris-HCl pH 7.5, 200 mM MES, and 1 M NaCl). Lysed cells were subjected to JESS-based immunoblotting with monoclonal CFTR-NBD2 antibody (1:100 dilution, #596, John R. Riordan and Tim Jensen, University of North Carolina, Chapel Hill, USA) through the Cystic Fibrosis Foundation Therapeutics Antibody Distribution Program (Bethesda, USA Company). Uncropped and unprocessed gel images are provided in the Source Data.

## Ribo-seq mapping and computing of translation speed

For calculating the A-site codon occupancy and deducing speed of translation, we used Ribo-seq (ribosome profiling) data from CFBE41o⁻ cells expressing WT-CFTR[40,57]. Sequenced reads were processed as follows: (i) quality selection using the *fastx-toolkit* (0.0.13.2) with a threshold of 20; (ii) adapter sequences removal by *cutadap* (1.8.3) with a minimal overlap of 1 nt; (iii) depletion of reads mapping to rRNA reference sequences (*bowtie* 1.2.2; -y –un); (iv) mapping the remaining reads to the human genome (GRCh38) using STAR[58] (2.5.4b) allowing maximum of one mismatch and filtering out reads mapping to multiple positions (--outFilterMismatchNmax 1 --outFilterMultimapNmax 1) to the longest annotated CDS for each transcript. Uniquely mapped reads were normalized to reads per million mapped reads (RPM) or reads per kilobase per million mapped reads (RPKM).

Ribosome-protected fragments (RPFs) from the Ribo-seq data set of CFBE41o⁻ cells were first calibrated to the ribosomal A site. The ribosomal dwelling occupancy at A-site codons was determined using the calibration tool (https://github.com/AlexanderBartholomaeus/MiMB_ribosome_profiling)[59]. Briefly, RPFs were binned into equal read-length groups and aligned to the start codon of each transcript. Using the 5′-most nucleotide of each read, we calculated the offset for each read length to the start codon in the P site. The A-site position was calculated by adding three nucleotides. To avoid initiation and termination bias through disproportional accumulation of reads, we excluded the first and last 17 codons of each transcript[40,60]. The higher the ribosomal dwelling occupancy of a given codon in the A site, the slower the rate of translation at that position and vice versa, with lower occupancy indicative of faster codons. By this approach, translation speed is proportional to the inverse of ribosomal occupancy and can be used as an estimate for the translation velocity of a codon[39,61–63]. Values for the 61 sense codons were used to compute translation velocity upstream of PTCs for *CFTR* and transcripts associated with PCD, mutations which confer potentially lethal pulmonary diseases. The translation velocity for the sequence (70 codons) upstream of each PTC was calculated by averaging the translation speed over a window of five codons.

## Modeling probability of ribosome collisions at PTCs

To determine the distribution of translation velocity across the human transcriptome, for each transcript we calculated the average speed of translation at every possible 5-codon window and related these values to the average translation speed of the respective transcript. The 66 codons adjacent to the start codon or upstream of the stop codon are translated with much slower velocity for regulatory purposes[38,64,65]. We considered only genuinely elongating ribosomes within the CDS. Under normal growth conditions, translating ribosomes are spatially well separated, with an average IRD between A sites of ~66 codons[38,42,44]. Thus, in translating regions, 66 codons from 5′ and 3′ of each CDS were excluded. In addition, transcripts shorter than 198 codons (2746 transcripts) were omitted from the analysis since they would represent transcripts with a single elongating ribosome. From the transcriptomic average speed, we used the value of 1.5 standard deviation ($\pm 1.5\sigma$) as a threshold (Supplementary Fig. 3a).

PTCs with segments for which subsequent changes of translation velocity above or below the $1.5\sigma$ threshold were selected and dependence on distance was classified as: close (5–35 codons), middle (20–45 codons) and distant (35–70 codons), according to the distance of change to the PTC. The probability of collision was computed using the ratio of the absolute translational speed ($|\Delta_{\text{speed}}|$) to the IRD (Eq. 1). The $\Delta_{\text{speed}}$ represents the difference between the translation velocity changes above and below $1.5\sigma$. The IRD is set by the distance between the PTC and the middle nucleotide of the segment with a velocity change:

$$Probability = \frac{|\Delta_{\text{speed}}|}{\text{IRD}} \qquad (1)$$

Collision probability landscape was next generated by applying the Eq. (1) using all possible $\Delta_{\text{speed}}$, values between 0 and 0.5 (with a step of 0.01), and IRD, between 10 and 100 (with a step of 0.1). The probability was adjusted between 0 and 0.5. tRNA concentration is well established as predictor of codon speed and inversely correlates with ribosome dwelling occupancy at an A site[40]. In eukaryotes, the difference between tRNAs with the highest and lowest concentration is ~10-fold[32,66]. Thus, the maximal difference in translation velocity over the threshold would be 0.5.

The probability score for each PTC was calculated and added to the collision probability landscape. Because average IRD between the A sites of two consecutive ribosomes of ~66 codons and between two colliding ribosomes is ten codons[38,42,44], we used 10–100 codons for this analysis.

## Reporting summary

Further information on research design is available in the Nature Portfolio Reporting Summary linked to this article.

## Data availability

The data supporting the findings of this study are available from the corresponding authors upon request. The Ribo-seq data generated in this study have been deposited within Gene Expression Omnibus under accession number GSE74365. Source data are provided with this paper.

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

## Acknowledgements

We thank Hillary Valley (Cystic Fibrosis Foundation laboratory; Lexington MA) for the 16HBEge cells. This research was supported by grants from the Cystic Fibrosis Foundation USA (IGNATO2010 to Z.I.), NIH (1R01HL136414-05 to Z.I. and E.J.S.), Mukoviszidose Institut gGmbH (Bonn), the research and development arm of the German Cystic Fibrosis Association Mukoviszidose e.V (2105 to S.A.), Hamburg Innovation C4T projects (C4T635 to Z.I. and S.A.), and the scholarship from ANII Uruguay (POS-EXT-2020-1-164944 to M.D.). We acknowledge financial support from the Open Access Publication Fund of Universität Hamburg.

## Author contributions

N.B., S.A., and Z.I. conceptualized the work and strategy. N.B., M.D., S.B., M.E., A.J.S., and S.A. performed the experiments and analyzed the data. N.B., L.S., and Z.I. conceptualized the model. L.S. developed the model and performed the analysis and simulations. A.R., E.J.S., and J.S.H. constructed the W1282X-CFTR and WT-CFTR Calu3 cell lines. N.B., L.S., M.D., E.J.S., S.A., and Z.I. analyzed the experiments and interpreted the data. Z.I. supported by N.B., L.S., S.A., and M.D. wrote the manuscript. All authors supported the review of the final version of the manuscript.

## Funding

## Competing interests

Z.I., S.A., N.B., and M.D. are inventors of patents related to tRNA designs for PTC correction. Z.I. is also a scientific advisor for Tevard Biosciences. E.J.S. is a non-voting trustee for the Cystic Fibrosis Foundation. The other authors declare no competing interests.
