## [Peer Review File · Nature Communications]

Translation velocity determines the suppression efficacy of pathogenic nonsense mutationsREVIEWER COMMENTS

Reviewer #1 (Remarks to the Author):

In this paper, the authors analyze various parameters affecting suppression of nonsense codons by tRNA suppressors. They show that the speed of translation upstream of the PTC affect suppression especially when the speed change occurs nearby within around 10 codons, because of collisions between ribosomes. The authors tested various PTCs and contexts with often two tRNA sup (given the number of observed mutations, an exhaustive and systematic analysis is out of reach) and reached a convincing conclusion with important consequences for our understanding of the molecular processes at play as well as for future therapeutic successes.

I have some questions that the authors may want to discuss.

I do not understand the use tRNASer; Ser is by far not the most prevalent amino acid impacted by the mutation (Arg and Gln or Glu are more frequent). They lose the advantage of tRNA sup to insert back the original amino acid. For tRNAArg, this is the case.

Do the authors have evidence that the tRNA sup transcripts are modified in vivo? If they are not modified, the efficiency of PTC reading by those tRNAs is very probably affected, rendering the competition with termination factors more critical (and the dependence on the translation speed). How long (and to what percentage) do they survive in vivo?

On page 3 towards the end, I do not understand the reference 23 for UAA being the most difficult PTC to be corrected.

Figure 5B is not easy to understand. Some descriptive explanations would be welcome. Same with Extended Data 5.

In Supplementary Table 1, there is I1127X, how is Isoleucine giving a stop codon? I guess that in N4487X, N is not Asn.

Reviewer #2 (Remarks to the Author):

Have you ever wondered if tRNA molecules could eventually be used as drugs? Over the past few years, Zoya Ignatova's team has been testing the use of tRNA molecules to cure genetic disorders caused by premature stop codons. And I am excited to see this paper describing their progress in understanding why some premature stop codons are more resistant to targeting with therapeutic tRNAs than others. They attribute this resistance to ribosome collisions, which somehow prevent tRNA molecules from being used for protein synthesis by the ribosomal A site.

I do not have any problems with their experimental system, although it is highly complex, as any experimental system is. It is unclear, for instance, whether the tRNA transcripts used for transfection of their three model cell lines differ in stability, transfection efficiency, aminoacylation levels, etc., or whether the reporter genes compared in this study differed in transcription noise, which is known to have a dramatic impact on mRNA stability and frequency of translation initiation (<https://pubmed.ncbi.nlm.nih.gov/36441824/>). I am sure some of this will be addressed in their future studies.

My only recommendation is for the authors to consider the following: Currently, many small molecules are known to regulate ribosome density on mRNAs by targeting the initiation of translation. Many of these molecules are used as potent anticancer drugs (<https://biosignaling.biomedcentral.com/articles/10.1186/s12964-020-00607-9>). Because ribosome density on mRNAs appears to be a major determinant of sup-tRNA activity, I leave it up to the authors to decide whether to mention some of the literature on translation initiation inhibitors in their discussion section (currently, the manuscript appears to have only one section—introduction—but this is likely due to some minor processing errors). Personally, I would not be surprised if some molecules that are currently being developed to suppress cancer by reducing translation initiation could also

enhance sup-tRNA activity to a greater extent than the eRF1 inhibitor SRI-41315.

Reviewer #3 (Remarks to the Author):

In this manuscript entitled "Translation velocity determines the suppression efficacy of pathogenic nonsense mutations", Ignatova and colleagues tested the efficacy of engineered tRNAs that can suppress the pathology by premature termination codons (PTC) in cultured cells. By using in vitro transcribed engineered suppressor tRNAs and PTC constructs, they showed that the efficiency of suppressor tRNAs varies in a PTC context- and cell type-dependent manner. By performing ribo-seq analysis, the authors found that deregulation of translation velocity depends on the sequence proximal to PTC. Lastly, they showed that ribosomal collisions proximal to PTC affected the suppression efficacy of tRNAs. The findings reported here provide very useful information regarding how translation velocity is regulated in PTC-containing transcripts, and how engineered tRNAs can be applied to suppress PTC-mediated pathogenesis. Following are my concerns that the authors should address to improve the paper.

Major comments:

1. I think the authors should examine whether suppressor tRNAs can alleviate disease-like phenotypes caused by expression of PTC variant proteins. For example, does ectopic expression of CFTR variants in CFBE41o- cells cause disease-like phenotypes? If so, I think the authors should also test whether this phenotypic change is suppressed by engineered suppressor tRNA treatment. Performing these experiments will be crucial to show potential application of engineered tRNAs for PTC-prone diseases.
2. In figure 3, the authors addressed that mRNA translation velocity modulates PTC suppression susceptibility. However, I think the data are not sufficient to support the statement. The author should test PTC suppression susceptibility by sup-tRNA under diverse mRNA translation velocity conditions.
3. In figure 4, they showed the possibility of the role of ribosomal collisions at PTCs with respect to the suppression efficacy, but the association is correlational. They need to test the causal effect by genetically inhibiting RQC factors (e.g. ZNF598 knockdown).
4. The authors need to add a discussion paragraph regarding the application of their findings in human diseases and testing the possibility in vivo using animal models.

Minor comments:

1. The title of this paper requires improvement. I think one of the keywords of this paper is the suppression efficacy of engineered tRNAs, but it is lacking in the current title.
2. The authors calculated the relative protein amounts and denoted normalized loaded protein amounts below each blot in figure 4. I think the labels are confusing and may provide misleading information to the readers. I would add quantification plots of GCN1 and ZNF598 for clearance.
3. In figure 1b, I would separate the plot displaying the readthrough efficiency of sup-tRNA^{SerUCA} decoding UGA at PTC mutations in RSPH4A, DNAH5, and CFTR (labeled in purple, blue, and green dot, respectively) by respective colors.
4. In figure 1b, please consider alternative ways to represent the statistical significance of the data, as the current one is confusing.

Point-by-point responses to the Referees' comments

In the revised manuscript, the changes in response to Reviewers' comments and suggestions are highlighted blue. The small orthographic and expression improvements at few places are not highlighted.

Reviewer #1:

In this paper, the authors analyze various parameters affecting suppression of nonsense codons by tRNA suppressors. They show that the speed of translation upstream of the PTC affect suppression especially when the speed change occurs nearby within around 10 codons, because of collisions between ribosomes. The authors tested various PTCs and contexts with often two tRNA sup (given the number of observed mutations, an exhaustive and systematic analysis is out of reach) and reached a convincing conclusion with important consequences for our understanding of the molecular processes at play as well as for future therapeutic successes. I have some questions that the authors may want to discuss.

We are delighted to read the positive feedback of the Reviewer. We also thank for their critical suggestions, which we have rigorously addressed, including some new experiments, as explained below.

I do not understand the use tRNA^{Ser}; Ser is by far not the most prevalent amino acid impacted by the mutation (Arg and Gln or Glu are more frequent). They lose the advantage of tRNA sup to insert back the original amino acid. For tRNA Arg, this is the case.

To address the effect of the sequence context, we reasoned that we should keep all other parameters the same, namely, use equally long fragments from cDNA (without introns), same sup-tRNA. In this way, we avoid intrinsic differences in processing and expression level of the PTC-containing mRNAs, or different efficacy of sup-tRNAs carrying different amino acid. As we elaborated in our previous publication (PMID 7258671), the physico-chemical properties of the amino acid the sup-tRNAs carry influence their efficacy as a suppressor. We selected one potent tRNA, which is sup-tRNA^{Ser} (PMID 7258671). In the majority of the experiments, we use a segment flanking the corresponding PTC fused to a reporter, thus the misincorporation of a serine does not affect the result. We believe this is a correct experimental strategy, namely to fix all variables in the system and leave as a changing parameter the sequence context.

Reflecting on the comment of the Reviewer, we believe that we had not well explained this issue. In the revised version, we elaborated on our reasoning of using a single sup-tRNA (p. 4, first subsection in the results "Short PTC sequence context affects sup-tRNA efficacy only marginally").

Do the authors have evidence that the tRNA sup transcripts are modified in vivo? If they are not modified, the efficiency of PTC reading by those tRNAs is very probably affected, rendering the competition with termination factors more critical (and the dependence on the translation speed).

IVT administered tRNAs are not modified. Modifications are mostly installed in the nucleus. We discuss this issue in our recent review (PMID: 38049504) and compare IVT tRNAs administered as lipid-nanoparticles directly in the cytosol to tRNAs expressed episomally, i.e. suitable for AAV-administrations. The administration type is tissue specific, i.e. lungs are not suitable for episomal AAV administration and IVT tRNA-LNP is the preferred constellation, while CNS is amenable for episomal AAV administration.

In our recent experimental publication (PMID 7258671), we show that the lack of modifications does not influence the sup-tRNA fidelity. We observed only the insertion of the

targeted amino acid. The unmodified tRNA is stable over 72h (PMID 7258671). Importantly, even without the modifications, we achieved a very high efficacy of suppression in mouse models (~75%) and restored disease protein over the clinically relevant threshold for cystic fibrosis (PMID 7258671). The T-arm is important for the formation of the ternary complex (aa-tRNA-eEF1A-GTP), and consequently, for the operational efficacy of each tRNA. We believe that the targeted modifications of the T-arm we implement in our design strengthen the interactions of sup-tRNA with eEF1A that boost the translation activity and thus, may antagonize NMD and outcompete eRF (PMID: 34158503; PMID 7258671). In addition, we note that natural tRNAs that are evolutionary shaped with similar fidelity are differently modified, some hypermodified tRNAs carry 17 modifications, but many are hypomodified carrying only few (3-4) modifications.

1. *How long (and to what percentage) do they survive in vivo?*

Please refer to the comment above. Briefly, in our previous publication we measured that in mouse and in patient-derived primary cells sup-tRNAs are stable for more than 72h (PMID 7258671).

2. *On page 3 towards the end, I do not understand the reference 23 for UAA being the most difficult PTC to be corrected.*

We apologize for the inappropriate citation. We have amended this (p. 4) and defined precisely the readthrough entities.

3. *Figure 5B is not easy to understand. Some descriptive explanations would be welcome. Same with Extended Data 5.*

We have included more explanations on the model and interpretation of the results (p. 12-13)

4. *In Supplementary Table 1, there is I1127X, how is Isoleucine giving a stop codon? I guess that in N4487X, N is not Asn.*

The Reviewer is correct that Ile cannot be directly mutated to a PTC. This mutation is insertion mutation 3378_3379insTG (p.Ile1127Ter). Similarly, N4487X is a duplication mutation c.13458dup (p.Asn4487Ter). We apologize for not specifically designating those. Because of their specific character that differs from a single SNP-induced PTCs, we have omitted them from the table.

Reviewer #2

Have you ever wondered if tRNA molecules could eventually be used as drugs? Over the past few years, Zoya Ignatova's team has been testing the use of tRNA molecules to cure genetic disorders caused by premature stop codons. And I am excited to see this paper describing their progress in understanding why some premature stop codons are more resistant to targeting with therapeutic tRNAs than others. They attribute this resistance to ribosome collisions, which somehow prevent tRNA molecules from being used for protein synthesis by the ribosomal A site.

We are delighted to read the positive feedback of the Reviewer. We are grateful for their suggestions and considered them in the Discussion part.

I do not have any problems with their experimental system, although it is highly complex, as any experimental system is. It is unclear, for instance, whether the tRNA transcripts used for transfection of their three model cell lines differ in stability, transfection efficiency, aminoacylation levels, etc., or

whether the reporter genes compared in this study differed in transcription noise, which is known to have a dramatic impact on mRNA stability and frequency of translation initiation (<https://pubmed.ncbi.nlm.nih.gov/36441824/>). I am sure some of this will be addressed in their future studies.

This is an excellent point, which opens up a whole next level of questions to be addressed to explain tissue-specificity of disease phenotype that also reflects the tissue-specificity of administration and correction we observe. For example, as we show for sup-tRNA^{Arg}, (PMID 7258671) the readthrough efficacy drops down from ~60% on cDNA in cell culture model, to ~25% when using full-length DNA with all exons and introns in CRISPR-edited pulmonary immortalized cells, reaching ~14% when moving to primary patient-derived nasal epithelia. Thereby, the 5'UTRs were the same. Along with the Reviewer's insightful query, we firmly believe that the concentration of translational pools, which significantly vary among cells and cell types in humans, are the primary determinant of suppression efficacy. Factors such as transcriptional noise and the differing ribosomal concentrations across cells contribute to variations in initiation rates. Additionally, the fluctuating tRNA pool in different cells exerts varying effects on translation speed.

Acknowledging Reviewer's comment, we have relocated the section briefly discussing these issues to the discussion part of the manuscript. We have expanded the discussion (p. 14-15) to encompass these factors, emphasizing their consideration in the ongoing development of the model to enhance its predictive power, particularly in a tissue-specific context.

My only recommendation is for the authors to consider the following: Currently, many small molecules are known to regulate ribosome density on mRNAs by targeting the initiation of translation. Many of these molecules are used as potent anticancer drugs (<https://biosignaling.biomedcentral.com/articles/10.1186/s12964-020-00607-9>). Because ribosome density on mRNAs appears to be a major determinant of sup-tRNA activity, I leave it up to the authors to decide whether to mention some of the literature on translation initiation inhibitors in their discussion section (currently, the manuscript appears to have only one section—introduction—but this is likely due to some minor processing errors). Personally, I would not be surprised if some molecules that are currently being developed to suppress cancer by reducing translation initiation could also enhance sup-tRNA activity to a greater extent than the eRF1 inhibitor SRI-41315. Personally, I would not be surprised if some molecules that are currently being developed to suppress cancer by reducing translation initiation could also enhance sup-tRNA activity to a greater extent than the eRF1 inhibitor SRI-41315.

We followed the suggestion of the Reviewer to extend the discussion with this possibility (please refer to p. 14-15). The use of initiation inhibitors – which we pursue, but screen for novel ones – would establish a combo therapy by nonsense mutations on which sup-tRNA efficacy does not cross the therapeutic threshold.

Reviewer #3:

In this manuscript entitled "Translation velocity determines the suppression efficacy of pathogenic nonsense mutations", Ignatova and colleagues tested the efficacy of engineered tRNAs that can suppress the pathology by premature termination codons (PTC) in cultured cells. By using in vitro transcribed engineered suppressor tRNAs and PTC constructs, they showed that the efficiency of suppressor tRNAs varies in a PTC context- and cell type-dependent manner. By performing ribo-seq analysis, the authors found that deregulation of translation velocity depends on the sequence proximal to PTC. Lastly, they showed that ribosomal collisions proximal to PTC affected the suppression efficacy of tRNAs. The findings reported here provide very useful information regarding how translation velocity is regulated in PTC-containing transcripts, and how engineered tRNAs can be

applied to suppress PTC-mediated pathogenesis. Following are my concerns that the authors should address to improve the paper.

We are pleased to read the assessment of the Reviewer who points out the strength of our study. We also thank for their constructive critique which we have thoroughly considered in our revision.

Major comments:

1. I think the authors should examine whether suppressor tRNAs can alleviate disease-like phenotypes caused by expression of PTC variant proteins. For example, does ectopic expression of CFTR variants in CFBE41o- cells cause disease-like phenotypes? If so, I think the authors should also test whether this phenotypic change is suppressed by engineered suppressor tRNA treatment. Performing these experiments will be crucial to show potential application of engineered tRNAs for PTC-prone diseases.

All those points the Reviewer raises, we presented in our recent publication (PMID 7258671). We present in vivo (in mouse) efficacy and in FDA-endorsed cell models for cystic fibrosis (CF), including CF patient-derived fibroblasts. Sup-tRNAs show high efficacy and safety in vivo, restored CFTR expression and augment function at a level exceeding the therapeutic threshold for CF disease, thus emphasizing the high therapeutic potential of sup-tRNAs. In this work, we noticed, however, the differences in efficacy of the same sup-tRNA at different PTC contexts – a question we mechanistically address in this paper. Reflecting on the Reviewer's comment, we have edited the introduction to elaborate on these previous findings (please see p. 3).

2. In figure 3, the authors addressed that mRNA translation velocity modulates PTC suppression susceptibility. However, I think the data are not sufficient to support the statement. The author should test PTC suppression susceptibility by sup-tRNA under diverse mRNA translation velocity conditions.

We have added new data to Fig. 3 (panel 3c) which shows how differences in mRNA translation velocity at different PTC sites affect the sup-tRNA efficacy. Along with the data in Fig. 4, in which we present data on modulating translation velocity upstream of the PTC by introducing synonymous codons, we conclude the causal link between translation speed in PTC vicinity and sup-tRNA efficiency. Please also refer to the explanations on p. 8.

3. In figure 4, they showed the possibility of the role of ribosomal collisions at PTCs with respect to the suppression efficacy, but the association is correlational. They need to test the causal effect by genetically inhibiting RQC factors (e.g. ZNF598 knockdown).

Recent literature on the mechanisms of ribosome collisions has identified ZNF598 as a reliable collision marker, enabling the differentiation between genuine collisions and spatially close disomes (PMID: 30293783; PMID: 32615089; PMID: 35040509; PMID: 35491909; PMID: 31981475). This feature serves as a valuable tool in our assay. This robust assay allows for the selection of PTCs with high probability of experiencing collisions compared to those without. It's important to note that our focus is not on studying the mechanisms of collisions at PTCs; instead, we leverage this marker to pinpoint collision-prone PTC sequence contexts. As a result, we believe the suggested experiment falls outside the scope of this work.

The scarcity of sense tRNA determines the frequency of natural collisions at vacant sense codons – a process also sensed by ZNF598. ZNF598 depletion leads to defective resolution of stalled ribosomes (PMID: 28132843), resulting in an elevation of general collisions (those collisions are also detected in our system, Fig. 4a, WT-CFTR expressing cells). Conducting an experiment following ZNF598 depletion, would not be able to differentiate collisions at PTCs from general background collisions at sense codons. In contrast, at PTCs, the

competition between sup-tRNA and eRF1 determines collision frequency. We are convinced that the experiment we introduced with inhibition of eRF1 (Fig. 5) is selective for collisions at PTCs untangling them from the broader landscape of collisions.

4. The authors need to add a discussion paragraph regarding the application of their findings in human diseases and testing the possibility in vivo using animal models.

As suggested by the Reviewer, we added a discussion on the feasibility of combination therapy with adjuvants. Please refer to p.9.

Minor comments:

1. The title of this paper requires improvement. I think one of the keywords of this paper is the suppression efficacy of engineered tRNAs, but it is lacking in the current title.

We have edited the title to: Translation velocity determines the efficacy of engineered suppressor tRNAs on pathogenic nonsense mutations

2. The authors calculated the relative protein amounts and denoted normalized loaded protein amounts below each blot in figure 4. I think the labels are confusing and may provide misleading information to the readers. I would add quantification plots of GCN1 and ZNF598 for clearance.

We agree with the Reviewer and edited Figure 5 (formerly Figure 4) to present more intuitive values, namely the normalized band intensity on the total loaded protein amount.

3. In figure 1b, I would separate the plot displaying the readthrough efficiency of sup-tRNA^{SerUCA} decoding UGA at PTC mutations in RSPH4A, DNAH5, and CFTR (labeled in purple, blue, and green dot, respectively) by respective colors.

We understand the comment of the Reviewer and left free space to visually separate PCD-associated mutations from those in CFTR. We did not present them in a separate plot or panel, as we would like to have all contexts separated by the PTC identity rather than by the disease they originate from. Also, in this particular assay, to decouple disease gene expression specificities (i.e. natural variations of expression levels of full-length proteins, mRNA stability etc), we use only fragments surrounding the PTC which allows comparisons independent of pathology. All PCD mutations are differently color coded than CFTR to designate their associations with different pathology.

4. In figure 1b, please consider alternative ways to represent the statistical significance of the data, as the current one is confusing.

We have followed the suggestion of the Reviewer and edited the asterisks to be exactly over the values whose statistical significance they represent.

REVIEWERS' COMMENTS

Reviewer #1 (Remarks to the Author):

The authors have adequately address the points I raised. I have a small comment concerning one of their responses that I find misplaced.

"In addition, we note that natural tRNAs that are evolutionary shaped with similar fidelity are differently modified, some hypermodified tRNAs carry 17 modifications, but many are hypomodified carrying only few (3-4) modifications."

The authors should know that not every single tRNA sequence has the same potential for folding in a state appropriate for aminoacylation or A-site binding. The number of modifications in the core of the tRNA and in the anticodon loop reflects this fact. It is also known that an increase in tRNA expression can alleviate the loss in biological efficacy due to a lack of modifications (by mass action).

Reviewer #2 (Remarks to the Author):

Thank you for addressing my concern in full. I can't wait to see these two research directions merge!

Sergey Melnikov

Reviewer #3 (Remarks to the Author):

The authors sufficiently addressed my concerns.

Point-by-point responses to the Referees' comments

We are pleased that our revisions have addressed all points raised by the Reviewers. We thank them for their constructive critique.

Reviewer #1:

The authors have adequately address the points I raised. I have a small comment concerning one of their responses that I find misplaced.

"In addition, we note that natural tRNAs that are evolutionary shaped with similar fidelity are differently modified, some hypermodified tRNAs carry 17 modifications, but many are hypomodified carrying only few (3-4) modifications."

The authors should know that not every single tRNA sequence has the same potential for folding in a state appropriate for aminoacylation or A-site binding. The number of modifications in the core of the tRNA and in the anticodon loop reflects this fact. It is also known that an increase in tRNA expression can alleviate the loss in biological efficacy due to a lack of modifications (by mass action).

This is a sentence we had in our response to comment #3 of the Reviewer. Indeed, it was mentioned to support the exact same line of thoughts of the Reviewer mentioned above. Indeed, we also believe that the diversity of the modifications by their number, chemical identity and degree of modification at each nucleotide, idiosyncratically shape each tRNA to uniquely equalize their efficacy in decoding.

Reviewer #2

Thank you for addressing my concern in full. I can't wait to see these two research directions merge!

Sergey Melnikov

Reviewer #3:

The authors sufficiently addressed my concerns.